# Coexistence of distinct skyrmion phases observed in hybrid ferromagnetic/ferrimagnetic multilayers

Andrada-Oana Mandru [1✉], Oğuz Yıldırım [1], Riccardo Tomasello [2], Paul Heistracher [3], Marcos Penedo [1], Anna Giordano[4], Dieter Suess[3], Giovanni Finocchio [4✉] & Hans Josef Hug[1,5✉]

Materials hosting magnetic skyrmions at room temperature could enable compact and energetically-efficient storage such as racetrack memories, where information is coded by the presence/absence of skyrmions forming a moving chain through the device. The skyrmion Hall effect leading to their annihilation at the racetrack edges can be suppressed, for example, by antiferromagnetically-coupled skyrmions. However, avoiding modifications of the inter-skyrmion distances remains challenging. As a solution, a chain of bits could also be encoded by two different solitons, such as a skyrmion and a chiral bobber, with the limitation that it has solely been realized in B20-type materials at low temperatures. Here, we demonstrate that a hybrid ferro/ferri/ferromagnetic multilayer system can host two distinct skyrmion phases at room temperature, namely tubular and partial skyrmions. Furthermore, the tubular skyrmion can be converted into a partial skyrmion. Such systems may serve as a platform for designing memory applications using distinct skyrmion types.

[1] Empa, Swiss Federal Laboratories for Materials Science and Technology, Dübendorf CH-8600, Switzerland. [2] Institute of Applied and Computational Mathematics, FORTH, Heraklion-Crete GR-70013, Greece. [3] Christian Doppler Laboratory for Advanced Magnetic Sensing and Materials, Faculty of Physics, University of Vienna, Boltzmanngasse 5, Vienna 1090, Austria. [4] Department of Mathematical and Computer Sciences, Physical Sciences and Earth Sciences, University of Messina, Messina I-98166, Italy. [5] Department of Physics, University of Basel, Basel CH-4056, Switzerland. ✉email: andrada-oana.mandru@empa.ch; giovanni.finocchio@unime.it; hans-josef.hug@empa.ch

**M**agnetic skyrmions[1,2] have been observed in ferromagnets[3–8], ferrimagnets[9,10], and synthetic antiferromagnets[11], and the underlying physics of their stabilization involves a delicate balance of different micromagnetic energies[8,12–15]. Essentially, the Dzyaloshinskii–Moriya interaction (DMI) breaks the symmetry of the exchange energy and drives the stabilization of chiral structures, such as skyrmions. These are characterized by an integer winding number and hence have peculiar topological properties[14,16], that can be used for new device functionalities and computing architectures[12–14,17–21], but also for energetically efficient storage such as racetrack memories[19–22]. Even though skyrmions have been identified as the ideal information carriers, there are two fundamental problems that prevent their successful implementation in a racetrack device. The first problem is not being able to restrict the skyrmion motion along straight paths, which may result in a loss of skyrmions at the device edge. One solution to this problem is the use of antiferromagnetically coupled skyrmions[11,23,24]. The second problem is not having stable/constant inter-skyrmion distances, which leads to not having a controlled distance among bits[25]. One potential solution was proposed by Müller as a two-lane racetrack device, in which the information is stored in the lane number of each skyrmion[26]. Zheng et al. revealed a potential solution to this second problem with the first experimental observation of coexisting chiral bobbers and skyrmions[27]. This finding opened the possibility that a chain of binary data bits could be encoded by the two different solitons.

Materials hosting two distinct skyrmion phases add an additional degree of freedom in designing devices with improved properties in the emerging field of skyrmionics[28–30]. Modeling work showed that such an approach could for example be used to lower error rates in racetrack memories relying on skyrmions as information carriers[31]. However, to date, the simultaneous existence of two different soliton phases, such as tubular skyrmions and chiral bobbers, has only been shown in B20-type FeGe single crystalline materials[27] and in FeGe grown on Si(111) substrates by molecular beam epitaxy[32] at low temperatures, thus hindering the efficacy for future applications. Although they show very interesting physics, such single crystalline systems restrict the choice of materials for additional layers needed in devices for example to provide spin–orbit torque, an exchange field or simply for electrons used for applying currents or measuring the skyrmion induced Hall signals[33].

In this work, we develop a thin film system by combining two ferromagnetic skyrmion (SK) layers separated by a ferrimagnetic (Fi) layer with interfacial DMI, designed to support two distinct skyrmion phases at room temperature that is in principle compatible with other layers needed for device functionality. Our quantitative magnetic force microscopy (MFM) analyses performed on a series of samples designed for disentangling the MFM contrast arising from different layers clearly show the stabilization of a tubular skyrmion, which is extended in all the ferromagnetic and ferrimagnetic layers, and a partial/incomplete skyrmion nucleated in the ferromagnetic layers only. The diameters of the tubular and incomplete skyrmions are noticeably different, thus making these configurations suitable for a direct electrical detection by measuring the magnetoresistive signal. The experimental findings are supported quantitatively by micromagnetic simulations, demonstrating that the key ingredient to achieve the coexistence of the two skyrmion phases is the design of a Fi layer with a sufficiently large DMI. This work paves the way for the development of hybrid systems by merging together a variety of properties from different materials to move the skyrmionics a step forward towards practical applications. In particular, this work can impact the design of new robust skyrmion-based memory and computing architectures working at room temperature and coding the information in two types of skyrmions.

## Results

**Sample information**. Our sample contains two 14.5-nm thick [Ir (1)/Fe(0.3)/Co(0.6)/Pt(1)]$_{\times 5}$-multilayers (SK layers) separated by a 3.8-nm thick Fi layer (SK/Fi/SK, see Fig. 1a).

Previous studies[6] demonstrated that the Ir/Fe[34] and Co/Pt[35] interfaces in the Ir/Fe/Co/Pt-multilayers exhibit a strong negative DMI supporting skyrmions with a clockwise Néel wall texture at room temperature. The composition and layer thickness of the Fi layer permit an independent adjustment of its anisotropy and magnetization by the Tb:Gd ratio and rare-earth to Co-layer thickness, respectively. Here, a 1:1 Tb:Gd ratio was selected to obtain a weak effective perpendicular anisotropy $K_{eff} = 336 \pm 23$ kJ/m$^3$, determined by magnetometry measurements (see Supplementary Fig. 1 of Supplementary note 1). Such a weak positive anisotropy supports perpendicular magnetization structures with domains having a size up to several tens of microns. The SK/Fi/SK sample grown here is designed in such a way that skyrmions generated by the top and bottom SK layers could enter the Fi layer stabilizing skyrmion-like tubular structures penetrating through all layers of the sample. Consequently, high-resolution MFM has been used to study the micromagnetic state of the sample and its evolution as a function of the applied field.

**Magnetic force microscopy results**. Figure 1b–h shows MFM frequency shift ($\Delta f$) data with a total range of 1.4 Hz acquired at room temperature in fields $\mu_0 H_z$ ranging from 0 to 177 mT. MFM data of the remanent state show a red/blue (up/down) maze domain pattern with about $\pm 0.7$ Hz of $\Delta f$-contrast (Fig. 1b). If the field is increased from zero to $\mu_0 H_z = 122$ mT (Fig. 1d), the red (up) domains expand and inside these, skyrmions become more pronounced. At a field of $\mu_0 H_z = 133$ mT (Fig. 1e), almost all stripe domains collapse and skyrmions generating two distinct MFM signal levels (appearing with dark blue and white color) are observed. If the field (applied antiparallel to the skyrmion cores) is increased from 133–177 mT (Fig. 1e–h), the MFM contrast of all skyrmions is reduced and some of them are annihilated. However, while the MFM signal of the weak (white) skyrmions becomes gradually smaller with increasing field, the contrast of the strong (dark blue) skyrmions drops to the level of the weak (white) skyrmions at some critical field value (161 or 177 mT) after an initial gradual contrast decay of their MFM signal at lower fields (Fig. 1i–l).

The MFM contrast generated by a skyrmion arises from the convolution of its stray field above the sample surface with the magnetic charge distribution of the MFM tip[36]. Consequently, the radius of a skyrmion in an MFM image is wider than that of its spin texture. A reduction of the MFM signal in increasing fields is therefore compatible with a reduction of the skyrmion radius (see Supplementary Fig. 2 of Supplementary note 2). The distinct drop of the MFM signal observed for the strong skyrmions at fields of 161 or 177 mT can however not be explained by a gradual field-driven reduction of the skyrmion radius, but is indicative of a field-induced switching of the skyrmion type occuring in the SK/Fi/SK sample (see also Supplementary note 6).

For such a seemingly complicated sample hosting two types of skyrmions, it is important to understand the contributions coming from different layers. In order to disentangle the MFM contrasts, a series of additional samples have been fabricated: three consisting of selected parts of the SK/Fi/SK sample, and one in which the Fi layer was replaced by a magnetically inactive Ta layer of the same thickness. These sample structures are depicted in Fig. 2a–e together with their corresponding MFM data

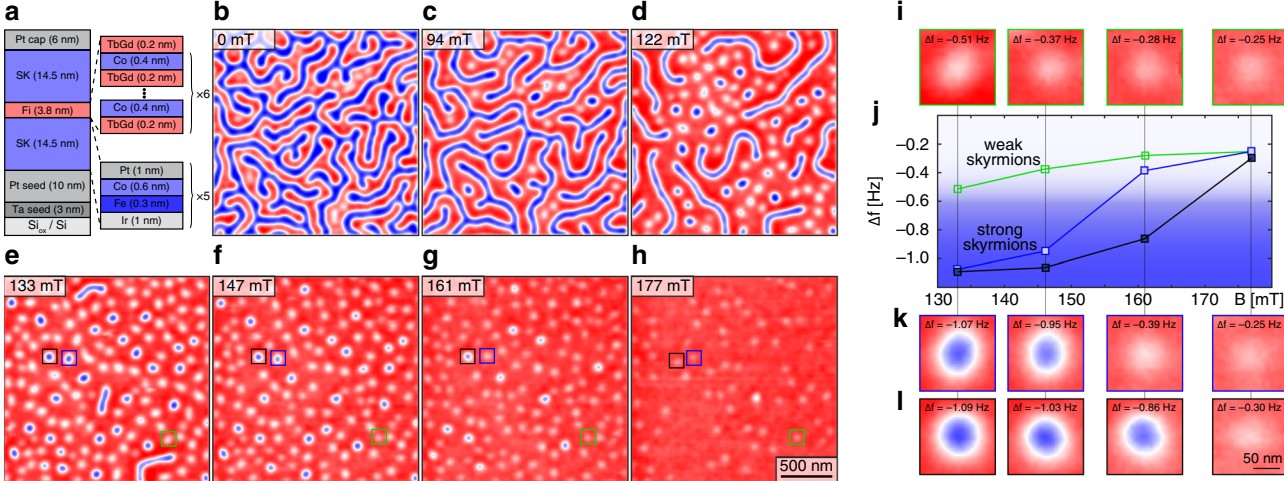

**Fig. 1 Magnetic force microscopy (MFM) results obtained on a trilayer sample supporting two different skyrmion phases. a** Schematics of the trilayer sample consisting of ferromagnetic top and bottom skyrmion (SK) layers, and a ferrimagnetic (Fi) layer. These three main layers again consists of a multilayer to achieve specific magnetic properties. **b–h** MFM data acquired in external fields $\mu_0 H_z = 0$–177 mT; note that the frequency shift ($\Delta f$) range for all MFM images is 1.4 Hz. Panels **i**, **k**, and **l** show magnified views of the skyrmions highlighted by the black, blue and green squares in panels **e–h** for fields of 133, 147, 161, and 177 mT. Panel **j** displays the evolution of the skyrmion contrast with the applied field.

(Fig. 2f–j), all acquired in a field of about 133 mT. Also visible are zoomed views of selected skyrmions, fitted Gaussian functions, and the fitted skyrmion contrast values (Fig. 2k–o). Note that in order to quantitatively compare the MFM signals obtained on different samples, a recently developed frequency modulated distance feedback method[37] was used. It permits keeping the tip-sample distance constant with a precision of about 0.5 nm over many days, even after reapproaching the tip on different samples and in applied magnetic fields. This can be achieved without ever bringing the tip in contact with the sample surface such that the magnetic coating of the tip remains intact and the same tip can be used for all samples.

For the sample consisting of a single SK layer ([Ir(1)/Fe(0.3)/Co(0.6)/Pt(1)]$_{\times 5}$ grown directly on Pt (Fig. 2a), the MFM data (Fig. 2f) reveals a disordered pattern of skyrmions with an areal density comparable to that observed in previous work[6] for similar samples with 20 Ir/Fe/Co/Pt repetitions. This confirms the existence of a strong negative DMI arising from the Ir/Fe[34] and Co/Pt[35] interfaces. The observed small variation of the MFM skyrmion contrast is attributed to variations of the local values of the DMI, perpendicular anisotropy and exchange stiffness[36]. The average skyrmion contrast $|\Delta f| = (0.32 \pm 0.05)$ Hz is obtained by fitting selected skyrmions (marked by the blue crosses) with 2D Gaussian functions. Zoomed MFM images of the skyrmions marked by the black squares, the fitted Gaussian functions, and the peak $\Delta f$ signal obtained from the fit are shown below Fig. 2f and in Fig. 2k. If the SK layer is grown directly onto the Fi layer (Fig. 2b), the $|\Delta f|$-contrast is reduced to $(0.24 \pm 0.02)$ Hz (Fig. 2g, l), but the areal density of the skyrmions remains about the same. Figure 2h shows MFM data acquired on a [Ir(1)/Fe(0.3)/Co(0.6)/Pt(1)]$_{\times 10}$-multilayer sample (Fig. 2c), representing two SK layers on top of each other. The observed skyrmion $|\Delta f|$-contrast in this case is $(0.71 \pm 0.05)$ Hz (Fig. 2m), slightly more than double the contrast observed for a sample consisting of a single SK layer (Fig. 2k). If the two SK layers are separated by a 3.8-nm thick Ta layer (Fig. 2d), the skyrmion $|\Delta f|$-contrast drops to $0.54 \pm 0.03$ Hz (Fig. 2i, n). Note that this contrast level agrees well to the $|\Delta f| = (0.58 \pm 0.06)$ Hz generated by the weak skyrmions in the SK/Fi/SK sample (Fig. 2e, j, o), where the two SK layers are separated by the Fi layer. However, the contrast remains much weaker than the $|\Delta f| = (1.05 \pm 0.09)$ Hz observed for the strong skyrmions in the

same sample. These observations suggest that the weak contrast arises from skyrmions existing solely in the bottom and top SK layers, whereas the strong skyrmion contrast is caused by a tubular skyrmion running through all three layers (see rectangles in Fig. 2e schematically representing the skyrmions). The much stronger contrast of the tubular skyrmions indicates that these spin textures must have a wider radius because their slightly increased length would not lead to the observed pronounced contrast increase.

**Micromagnetic simulations.** The physics behind the experimental stabilization of the different skyrmion spin textures is captured by 3-dimensional micromagnetic calculations and the results are summarized in Fig. 3 (see Methods and Supplementary Table 1 in Supplementary note 1 for the parameters used). Figure 3a shows an initial state with a clockwise skyrmion spin texture in the bottom and top SK layers and a uniform up magnetization state in the Fi layer (initial state 1). After a relaxation process and for a DMI of the Fi layer, $D_{Fi} = +0.8$ mJ/m$^2$, a clockwise skyrmion spin texture with a slightly larger radius appears in the four bottom-most Fe/Co-sublayers, B1 to B4 of the bottom SK layer, as well as in all Fe/Co-sublayers, T1 to T5 of the top SK layer (final state 1, see Fig. 3b). As also visible from the cross-section in Fig. 3c, the skyrmion diameter is thickness dependent, being larger near the middle of the sample and smaller in the external layers (Fig. 3c), as expected from the minimization of the magnetostatic energy, and already observed for other simpler structures[8,15] (see Supplementary Fig. 3a in Supplementary note 3 for the dependence of the skyrmion radius on the layer position). The chirality is the same in all the layers, as expected from the energy minimization when the DMI is large enough with respect to the magnetostatic energy[8,15] (the DMI of the SK layer is $-2.5$ mJ/m$^2$). The magnetization of the Fi layer remains uniform and the ferromagnetic interlayer exchange coupling (IEC) through the 1 nm-thick Pt also suppresses the initial skyrmion in the top-most Fe/Co-sublayer, B5, of the bottom SK layer. Therefore, an incomplete skyrmion is obtained.

If a clockwise initial skyrmion spin texture is also imposed within the Fi layer (initial state 2, displayed in Fig. 3d), more complex spin textures develop provided $|D_{Fi}| > 0.7$ mJ/m$^2$ (see also Supplementary Fig. 3b in Supplementary note 3). The final

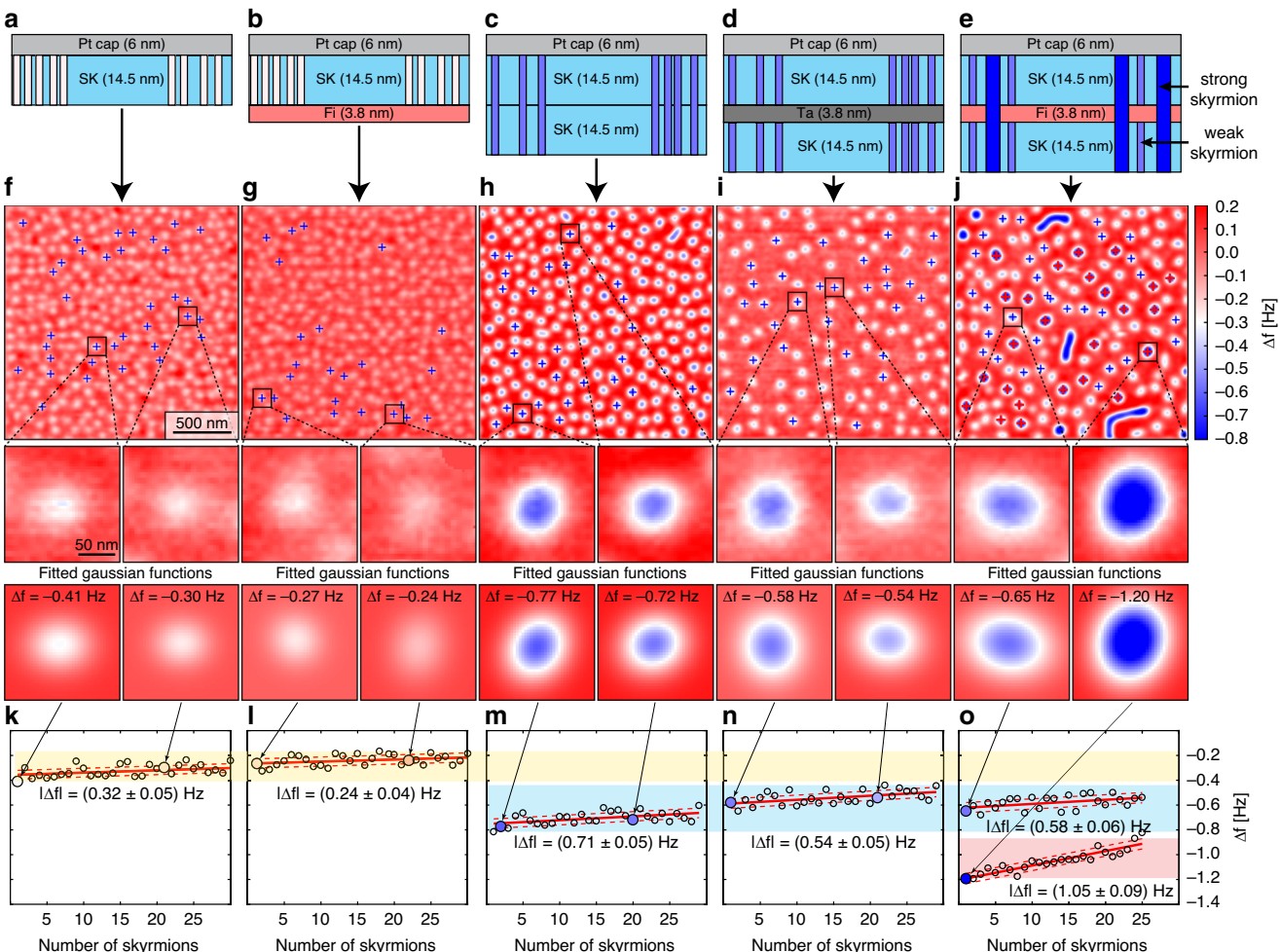

**Fig. 2 Different samples used to disentangle the contributions to the measured magnetic force microscopy (MFM) contrast on the skyrmion/ferrimagnet/skyrmion (SK/Fi/SK) sample. a** A single 14.5-nm thick ferromagnetic layer (SK) is deposited directly on the Pt-seed (not shown) on top of the oxidized Si-wafer. The vertical, white rectangles schematically depict the skyrmions. **b** The same ferromagnetic layer as in **a** but deposited on the $[(TbGd)(0.2)/Co(0.4)]_{\times 6}/(TbGd)(0.2)$ ferrimagnetic layer. **c** A sample consisting of two 14.5-nm thick SK layers. **d** Similar to **c** but the two SK layers are separated by a magnetically inactive 3.8-nm thick Ta layer. The skyrmions within the two SK layers are depicted by the vertical pale blue rectangles. **e** Schematics of the trilayer sample consisting of two SK layers separated by a 3.8-nm thick Fi layer; the skyrmions are depicted by the vertical pale blue rectangles penetrating through the SK layer and by the wider dark blue rectangles for the strong contrast skyrmions penetrating through all layers. **f–j** 2 × 2 μm MFM data (top row) and zoomed images of skyrmions shown together with the fitted Gaussian functions (lower rows) obtained on each of the samples in **a–e**. The crosses in panels **f–j** highlight the skyrmions which have been fitted to determine the frequency shift ($\Delta f$) contrast displayed in panels **k–o**. All images have been taken in a field of about 133 mT.

state 2 and its corresponding cross-sectional view are shown in Fig. 3e, f respectively, for $D_{Fi} = 0.8$ mJ/m². Differently from final state 1, skyrmions exist in all sublayers of the bottom and top SK layers, and also in the Fi layer. Hence, a tubular skyrmion with a larger radius is stabilized. Interestingly, the skyrmion chirality is thickness dependent. In the bottom-most Fe/Co-sublayer of the bottom SK layer, a Néel skyrmion with a counter-clockwise chirality opposite to the one favored by a negative DMI is stabilized. This chirality leads to an improved magnetic flux closure and hence optimizes the magnetostatic energy. Layers B2 up to B4 have the clockwise chirality expected for the negative DMI of the SK layers[34], whereas the skyrmion chiralities in B5 and Fi layers derive from the trade-off among negative DMI of the SK layer, positive $D_{Fi}$ and IEC. The skyrmion chirality in the Fi layer is intermediate between Néel outward and Bloch types, thus reminiscent to a counter-clockwise skyrmion expected for the positive $D_{Fi}$[35]. The frustration arising from the negative DMI in the SK layer, positive $D_{Fi}$ and IEC then explains the Bloch-type skyrmion obtained in layer B5, that compromises between

clockwise and counter-clockwise chiralities. Furthermore, this frustration also decreases slightly the skyrmion diameter in both the Fi and B5 layers (see also Supplementary Fig. 4 in Supplementary note 3 and compare the skyrmion diameter for positive and negative values of $D_{Fi}$). A simulation capturing the coexistence of tubular and incomplete skyrmions and its comparison to the MFM data are presented in Supplementary note 4. Also, for further analysis on the MFM image shown in Fig. 1f, see Supplementary note 5.

## Discussion

Since the SK/Fi/SK system presented here consists of multilayers with specifically tailored magnetic properties, it is possible that small variations in the properties of the individual layers could affect the tubular-to-incomplete skyrmion ratio. In fact, we already established that only particular values of $D_{Fi}$ attributed to the Fi layer give rise to tubular skyrmions, and therefore one may speculate that additional parameters, such as Fi anisotropy and

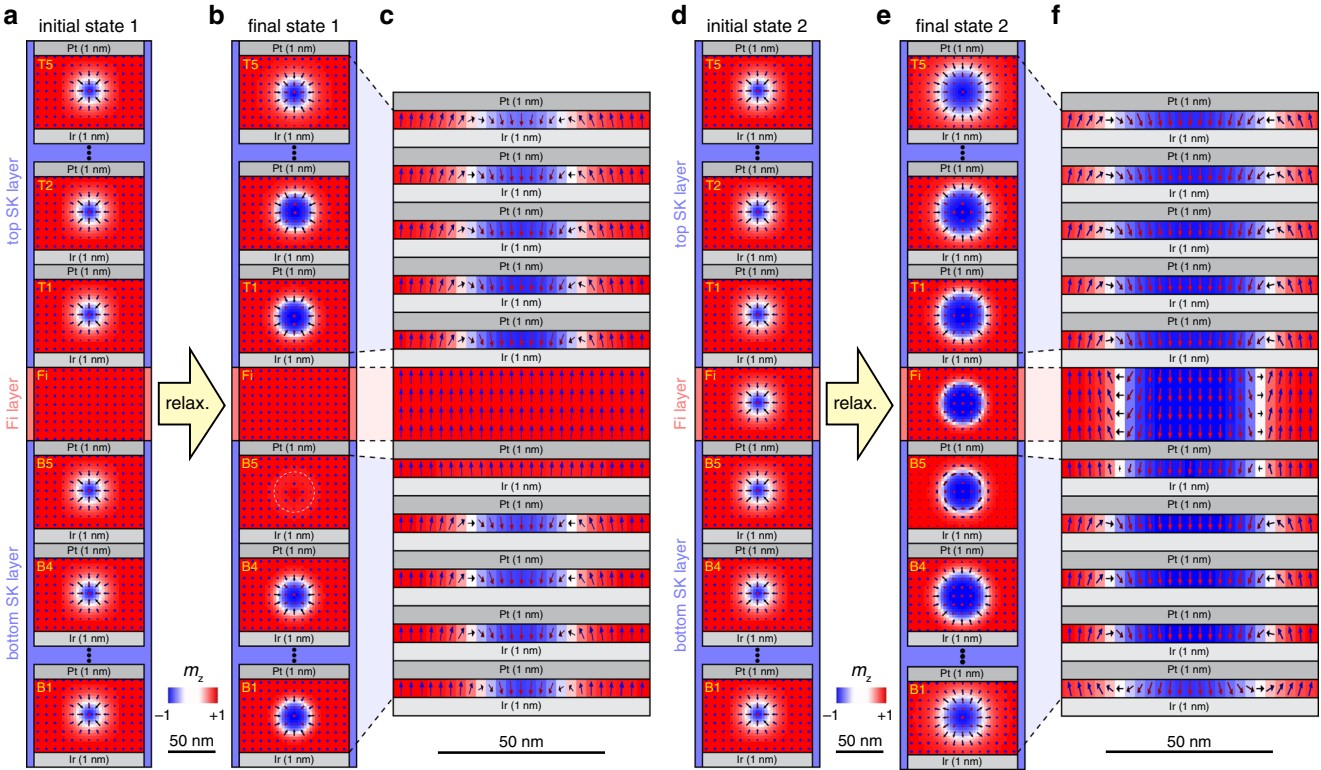

**Fig. 3 Micromagnetic simulations for the incomplete and tubular slyrmion phases. a** Intial state 1 with a clockwise skyrmion spin texture imposed in the bottom and top skyrmion (SK) layers but not in the ferrimagnetic (Fi) layer. **b, c** Final state 1 and its corresponding cross-section after relaxation of the initial state 1. A skyrmion spin texture exists solely in the sublayers B1 to B4 of the bottom SK layer and in all sublayers of the top SK layer, but not in the Fi layer. **d** Initial state 2 with a skyrmion spin texture enforced in all layers. **e, f** Final state 2 and its corresponding cross-section: a tubular skyrmion running through all layers is stabilized. The colors refer to the z-component of the normalized magnetization (blue-negative and red-positive).

layer thickness, could play a role in determining this ratio. Also, regarding the skyrmion generator layer, $[Ir(1)/Fe(0.3)/Co(0.6)/Pt(1)]_{\times 5}$, it was already established that changing the thickness of the ferromagnetic layers has a substantial influence on the skyrmion density[6]. This could be an alternative route to further tune the density of the two types of skyrmions.

In summary, the hybrid ferro/ferri/ferromagnetic multilayer system presented here supports the coexistence of two skyrmion phases at room temperature: a smaller-diameter incomplete skyrmions existing solely in the top and bottom ferromagnetic layers, and larger-diameter tubular skyrmions running through the entire sample. In future devices this may facilitate the electrical detection of the skyrmions and the distinction between the two states. Moreover, our metallic multilayer system permits the future implementation of additional layers, e.g., to generate a strong-spin–orbit torque, or layers providing an RKKY-type exchange to a layer with a perpendicular magnetization to allow the existence of skyrmions at zero field. The concept discussed here thus paves the way for future skyrmionic devices potentially including systems permitting the storage of information along the third dimension.

## Methods

**Magnetic force microscopy measurements and analysis**. The MFM measurements were performed using a home-built high-vacuum ($\approx 10^{-6}$ mbar) MFM system equipped with an in-situ magnetic field of up to $\approx 300$ mT. By operating the MFM in vacuum, we obtain a mechanical quality factor $Q$ for the cantilever of $\approx 200{,}000$. Using such high $Q$ values improves the sensitivity by a factor of about 40 compared to MFM performed in air and also permits the use of a thin magnetic coating on the tip to minimze the influence of its stray field on the micromagnetic state of the sample. SS-ISC cantilevers from Team Nanotech GmbH with a tip radius below 5 nm (without any initial coating) were used. In order to make the cantilever tip sensitive to magnetic fields, we sputter-deposited at room

temperature 3 nm of Co on a Ta seed (2 nm) and then capped with Ta (4 nm) to prevent oxidation. A Zurich Instruments phase locked loop (PLL) system was used to oscillate the cantilever on resonance at a constant amplitude of 7 nm and to measure the frequency shift arising from the tip-sample interaction force derivative. Note that the frequency shift is negative for an attractive force (derivative). For the MFM data shown in Figs. 1 and 2, an up field was applied and an MFM tip with an up magnetization was used. Therefore, the skyrmions have a down magnetization as those in our micromagnetic simulations (Fig. 3). The up tip magnetization and the down magnetization of the skyrmions then generates a positive frequency shift contrast that would correspond to a red color in the MFM images displayed in Figs. 1 and 2. In order to facilitate the comparison of the skyrmions measured by MFM with those obtained from micromagnetic calculations, the MFM data from Figs. 1 and 2 have been inverted.

**Sample preparation**. Samples were grown using DC magnetron sputtering under a 2 μbar Ar atmosphere using an an AJA Orion system with base pressure of $\approx 1 \times 10^{-9}$ mbar. All multilayers were deposited onto thermally oxidized Si (100) substrates with Ta(3 nm)/Pt(10 nm) as seed layers and Pt(6 nm) as capping layer (for oxidation protection). The substrates were annealed at $\approx 100$ °C for an hour and cooled down close to room temperature before each deposition. The layer thickness was determined by calibrations performed using X-ray reflectivity on samples containing single layers of each individual element. Since the single SK layer sample is very sensitive to the Fe and Co thicknesses, the reproducibility was verified periodically by re-growing such a sample and performing MFM measurements under the same conditions.

**Magnetometry measurements**. The bulk magnetic properties of the samples were determined[38] by vibrating sample magnetometry (VSM) using a 7 T Quantum Design system. The measurements were performed at 300 K for both in-plane and out-of-plane geometries and in fields of up to 4 T. All samples were measured using the same VSM holder and each measurement was repeated several times. In addition, the background signal coming from the VSM holder and bare substrate was periodically checked to ensure a clean magnetic signal coming from the ferro- and/or ferrimagnetic layers only.

**Micromagnetic simulations details**. The micromagnetic computations were carried out by means of a state-of-the-art micromagnetic solver, PETASPIN[39] and

magnum.af [40], both based on the finite difference scheme and which numerically integrate the Landau–Lifshitz–Gilbert (LLG) equation by applying the Adams–Bashforth time solver scheme:

$$\frac{d\mathbf{m}}{d\tau} = -(\mathbf{m} \times \mathbf{h}_{\text{eff}}) + \alpha_{\text{G}}\left(\mathbf{m} \times \frac{d\mathbf{m}}{d\tau}\right) , \quad (1)$$

where $\alpha_{\text{G}}$ is the Gilbert damping, $\mathbf{m} = \mathbf{M}/M_s$ is the normalized magnetization, and $\tau = \gamma_0 M_s t$ is the dimensionless time, with $\gamma_0$ being the gyromagnetic ratio, and $M_s$ the saturation magnetization. $\mathbf{h}_{\text{eff}}$ is the normalized effective field in units of $M_s$, which includes the exchange, interfacial DMI, magnetostatic, anisotropy and external fields[8,41]. The DMI is implemented as

$$\epsilon_{\text{InterDMI}} = D[m_z \nabla \cdot \mathbf{m} - (\mathbf{m} \cdot \nabla)m_z] . \quad (2)$$

The [Ir(1 nm)/Fe(0.3 nm)/Co(0.6 nm)/Pt(1 nm)]$_{\times 5}$ SK layers are simulated by five repetitions of a 1-nm thick CoFe ferromagnet separated by a 2-nm thick Ir/Pt non-magnetic layer. Each ferromagnetic layer is coupled to the other ones by means of the magnetostatic field only (exchange decoupled); for simplicity, we neglect any Ruderman–Kittel–Kasuya–Yosida (RKKY) interactions. For the SK layer, we used the following physical parameters: saturation magnetization $M_s = 1371 \pm 41$ kA/m, and uniaxial perpendicular anisotropy constant $K_u = 1316 \pm 92$ kJ/m$^3$ (both obtained by our VSM measurements), exchange constant $A = 15$ pJ/m, and interfacial DMI constant $D = -2.5$ mJ/m$^2$ from[6]. The ferrimagnetic [(TbGd)(0.2 nm)/Co(0.4 nm)]$_6$/(TbGd)(0.2 nm)-multilayers simulated by a 4 nm magnetic layer. Its saturation magnetization $M_{s,\text{Fi}} = 488 \pm 34$ kA/m, equal to the net magnetization of the experimental ferrimagnet, and its uniaxial perpendicular anisotropy constant $K_{u,\text{Fi}} = 486 \pm 44$ kJ/m$^3$ were again measured by VSM. The exchange constant $A_{\text{Fi}} = 4$ pJ/m was used in agreement with our prior work for rare-earth-transition metal alloy layers[42]. We use a discretization cell size of $3 \times 3 \times 1$ nm$^3$. The top ferromagnetic layer (B5) of the bottom SK layer is coupled to the first 1 nm of the ferrimagnetic layer via an RKKY-like interlayer exchange coupling[43]. The bottom Ir/Fe/Co/Pt layer finishes with a 1 nm-thick Pt layer that is known to lead to a large RKKY exchange coupling to the Fi layer; we set a positive value of the constant (ferromagnetic coupling) equal to 0.8 mJ/m$^2$ from[44]. The top SK layer grown on the top of the Fi layer starts with a 1 nm-thick Ir layer. Since the RKKY coupling through 1 nm of Ir is known to be very weak[45], it has been neglected here. In all the simulations, an out-of-plane external field $H_{\text{ext}} = 130$ mT is applied antiparallel to the skyrmion core.

## Data availability

All relevant data supporting this study are available from the authors (A.-O.M., H.J.H. and G.F.) upon reasonable request.

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

## Acknowledgements
A.-O.M., O.Y., and H.J.H. thank the Swiss National Science Foundation under Projects 200021-147084, 200021E-160637, 154410, and Empa for the financial support. R.T. and G.F. thank the project ThunderSKY funded from the Hellenic Foundation for Research and Innovation (HFRI) and the General Secretariat for Research and Technology (GSRT) under Grant No. 871 and the support by PETASPIN association. D.S. acknowledges the Austrian Science Fund under Grant I2214-N20 for financial support.

## Author contributions
H.J.H. and G.F. conceived the idea and planned the multidisciplinary approach to develop the project. H.J.H., A.-O.M., and O.Y. designed the multilayer systems. O.Y. and A.-O.M. grew the samples based on earlier samples grown and measured by M.P.; A.-O.M. performed the MFM experiments. O.Y. carried out the VSM measurements. H.J.H. performed the MFM data analysis. R.T. performed the micromagnetic simulations. A.G. developed the numerical methods and the software to simulate hybrid ferromagnetic/ferrimagnetic multilayer systems. R.T. and G.F. analyzed and interpreted the micromagnetic data. D.S. and P.H. performed micromagnetic calculations to obtain the stray field above the sample. All authors contributed to the writing of the manuscript.

## Competing interests
The authors declare no competing interests.
