## [Peer Review File · Nature Communications]

Reviewers' Comments:

Reviewer #1:

Remarks to the Author:

Mandru and coworkers report an experimental and simulation study of skyrmion tube states in hybrid ferromagnet/ferrimagnet/ferromagnet multilayers at room temperature. By using the magnetic force microscopy (MFM) technique, the authors find two different skyrmion tube states in the multilayer, which can coexist in the sample at certain out-of-plane magnetic field. The authors also study the possible spin configurations of the two different skyrmion tube states using their own micromagnetic simulators. The authors point out that the coexistence of different skyrmion tube states in the hybrid multilayers could be useful for future spintronic applications, such as the magnetic racetrack-type memory where binary bits are encoded by two different topological solitons.

The topological spin texture is an active topic in the field of magnetism and spintronics due to their potential applications in information storage, logic computing and even neuromorphic computing. The most promising topological spin texture is the so-called skyrmion object, which can be used for carrying information and can be manipulated by different methods including electric field and current. Early studies on skyrmions have focused on their static and dynamic properties in ultra-thin films and bulk materials. Recent years, the skyrmion community is tremendously interested in the skyrmion physics in multilayer films due to, for example, increased skyrmion stability and non-trivial dynamics driven by spin-orbit torques. This work demonstrates that different skyrmion tube structures can be generated and stabilized in ferromagnet/ferrimagnet/ferromagnet multilayers, which provides a way for the design of multilayers hosting different topological spin textures. The manuscript is very well written and easy to follow. The experimental data are of high quality and the interpretation is reasonable. For these reasons, I believe this work is an important contribution to the field and is of particular interest to researchers studying skyrmions in magnetic multilayers. Therefore, I would recommend publication of this manuscript provided that the following issues are addressed.

Major Comments

1. The authors find two distinct MFM signal levels, which are assumed to be generated by weak and strong skyrmion tubes, respectively. The weak skyrmion tube is actually two coupled skyrmion tubes in the top and bottom SK layers. The strong skyrmion is a skyrmion tube running through the SK layers and the Fi layer. I am interested in whether there is a similar phenomenon for the stripe domains. In Fig. 1, the stripe domains are all of strong signal level. Do the authors find stripe domains showing weak signal (or hybrid weak and strong signals)?
2. The authors demonstrate the coexistence of two distinct skyrmion tubes in the same sample. However, I am feeling the density of strong skyrmion is much smaller than that of weak skyrmion. What is the reason? Is it possible to control the density of strong skyrmion?
3. By increasing the out-of-plane magnetic field, the authors demonstrate the conversion from strong skyrmion to weak skyrmion. I am thinking the weak skyrmion may not collapse suddenly at a critical field when further increasing the magnetic field. Instead, it may convert to a weaker one before its annihilation due to the field-induced switching of sub-SK layers. I suggest the authors discuss this problem and I would like to see more MFM data at larger magnetic field.
4. Related to my above comment, the authors only study the relaxed spin configurations of the strong and weak skyrmion tubes. I believe the authors are able to perform a more complex but useful simulation for the purpose of understanding the field-induced transformation between strong skyrmion and weak skyrmion. Namely, I am interested in what will happen if a larger out-of-plane magnetic field is applied to the relaxed state given in Fig. 3f. I expect it will convert to the state given in Fig. 3c but with slightly smaller skyrmion diameter. However, there is a possibility

that the magnetic field switches the sub-SK layers instead of the Fi layer.

5. How do the authors make sure the weak skyrmion tubes are always coupled through the Fi layer? Why there is no decoupled weak skyrmion? Also, why there is no half weak skyrmion that only exists in the top or bottom SK layer? Are these problems related to the strength of magnetic field?

6. I do not understand why the authors only consider the interlayer exchange interaction between the Fi layer and B5 layer. What makes B5 layer different from T1 layer? Pt and Ir interfaces?

7. According to the simulation results, the chirality and helicity of sub-layer skyrmions are not uniform due to the strong magnetostatic interaction. As the current-induced skyrmion dynamics is jointly determined by the skyrmion structure (e.g. helicity), I am unsure the weak and strong skyrmion tubes could be dynamic stable when spin-orbit torques are applied to drive them. I suggest the authors comment on this point and emphasize all advantages of skyrmion tubes in the main text.

Minor Comments

1. Line 16: the racetrack-type memory device was first proposed by Parkin et al. as a domain-wall-based application. I suggest the authors cite the seminal paper on racetrack memory.

2. Line 20: regarding the antiferromagnetically coupled skyrmion in synthetic antiferromagnets, I suggest the authors cite [Nature Communications 7, 10293 (2016)] and [Nature Communications 10, 5153 (2019)] along with Ref. 8. The suppression of skyrmion Hall effect was studied in the two papers, while Ref. 8 mainly focuses on the static stabilization of nanoscale synthetic antiferromagnetic skyrmions.

3. Line 34: At the beginning of the introduction, I think it would be better to mention the first theoretical prediction of the existence of skyrmions in magnets and cite the seminal paper by Bogdanov et al.

4. Line 45: The simultaneous existence of different skyrmion states can also be realized in frustrated magnets, as numerically shown in [Nature Communications 6, 8275 (2015); Nature Communications 8, 1717 (2017); Nature Communications 8, 14394 (2017)]. Besides, strictly speaking the chiral bobber is a three-dimensional structure different to the skyrmion tube. It is actually not a skyrmion-like object, so it is incorrect to say it a skyrmion bobber.

5. Line 46: Please spell out MBE.

6. Line 129: Fig. 1f should be Fig. 2f.

7. Line 148: Please spell out 3D.

8. Line 184: $D > 0$ should be $D < 0$.

9. Line 242: A full stop is missing after the equation.

10. Figure 1: the orange square is not easily visible; I suggest the authors change the color to green to increase the readability. Figure 1: Please add color bar for the MFM data. Figure 2: The 'skyrmion number' usually means the topological charge of a skyrmion in the skyrmion community. How about change it to 'number of skyrmions'? Figure 3: It would be better to indicate the skyrmion diameter or simulated sample size.

Reviewer #2:

Remarks to the Author:

Mandru et al., introduce a new material architecture to stabilize magnetic skyrmions at room temperature.

Namely, a hybrid ferro/ferri/ferromagnetic multilayer which they argue can host two distinct skyrmion phases. Key to the coexistence of two skyrmion phases is the ferrimagnetic layer.

The purpose of their material is for the skyrmions generated by the top and bottom skyrmion layers to enter the ferrimagnetic layer. Thus, stabilize skyrmion-like tubular structures which penetrate through all layers of the sample.

Central to the study is also magnetic force microscopy measurements and associated quantitative analysis, which the team of Han Hug developed over the years.

Indeed, two magnetic force microscopy "contrasts" are observed with distinct field dependence. Following tests on several tailored samples, data suggest that one of two contrasts in magnetic force microscopy images arises from skyrmions existing solely in the bottom and top skyrmion layers, whereas the other is caused by tubular skyrmions running through all layers.

Experiments are complemented by micromagnetic simulations, giving details on the size and chirality of the textures across the multilayer. The simulations add credence to the proposal that the hybrid ferro/ferri/ferromagnetic multilayer favors the coexistence of two skyrmion phases at room temperature. That is, skyrmions at the top and bottom ferromagnetic layers and tubular skyrmions running through the entire sample.

Importantly, their material structure allows tunability of the skyrmion textures namely, tubular skyrmions can be converted into partial skyrmions.

I enjoyed reading this article. It is well written and easy to follow. The authors present a logical sequence of rationale and implementation. Their material structure introduces a novel tool to engineer (perhaps multiple) chiral textures.

The measurements are of high quality and the analysis of magnetic force microscopy is carefully done. I believe the authors know very well the limitations of magnetic force microscopy in elucidating details on skyrmion textures. The numerical studies accompanying the experiments offer the much needed support towards the key find of this work. Namely, two distinct type of skyrmion structures across the multilayer.

Although one may challenge the details of the interpretation provided by the authors, the experimental data are indicative of two magnetic textures.

Also, based on earlier studies on similar materials these textures must be magnetic skyrmions. The "contrast" characteristics the authors report can in fact be seen in other multilayers though not appreciated by the community before.

Hence, at the very least the authors deliver a new material and measurement platform and protocol to develop and study the coexistence of distinct "surface" and "bulk" magnetic skyrmions.

The simulations they report are realistic and within the norm and the results in tandem with the microscopy experiments. They help develop an interesting and potentially very useful protocol.

The work certainly encourages further studies in this field of research with promising technological impact as already mentioned their article.

I recommend publication.

Reply letter for Nature Communications manuscript NCOMMS-20-29411-T

Reviewer Comments:

Reviewer #1 (Remarks to the Author):

Mandru and coworkers report an experimental and simulation study of skyrmion tube states in hybrid ferromagnet/ferrimagnet/ferromagnet multilayers at room temperature. By using the magnetic force microscopy (MFM) technique, the authors find two different skyrmion tube states in the multilayer, which can coexist in the sample at certain out-of-plane magnetic field. The authors also study the possible spin configurations of the two different skyrmion tube states using their own micromagnetic simulators. The authors point out that the coexistence of different skyrmion tube states in the hybrid multilayers could be useful for future spintronic applications, such as the magnetic racetrack-type memory where binary bits are encoded by two different topological solitons.

The topological spin texture is an active topic in the field of magnetism and spintronics due to their potential applications in information storage, logic computing and even neuromorphic computing. The most promising topological spin texture is the so-called skyrmion object, which can be used for carrying information and can be manipulated by different methods including electric field and current. Early studies on skyrmions have focused on their static and dynamic properties in ultra-thin films and bulk materials. Recent years, the skyrmion community is tremendously interested in the skyrmion physics in multilayer films due to, for example, increased skyrmion stability and non-trivial dynamics driven by spin-orbit torques. This work demonstrates that different skyrmion tube structures can be generated and stabilized in ferromagnet/ferrimagnet/ferromagnet multilayers, which provides a way for the design of multilayers hosting different topological spin textures. The manuscript is very well written and easy to follow. The experimental data are of high quality and the interpretation is reasonable. For these reasons, I believe this work is an important contribution to the field and is of particular interest to researchers studying skyrmions in magnetic multilayers. Therefore, I would recommend publication of this manuscript provided that the following issues are addressed.

Reply: We thank the reviewer very much for the positive assessment of our work. We also appreciate the very careful reading of the manuscript, the useful comments, and the suggestions for improvement. Below we address in detail each comment point by point.

Major Comments

1. The authors find two distinct MFM signal levels, which are assumed to be generated by weak and strong skyrmion tubes, respectively. The weak skyrmion tube is actually two coupled skyrmion tubes in the top and bottom SK layers. The strong skyrmion is a skyrmion tube running through the SK layers and the Fi layer. I am interested in whether there is a similar phenomenon for the stripe domains. In Fig. 1, the stripe domains are all of strong signal level. Do the authors find stripe domains showing weak signal (or hybrid weak and strong signals)?

Reply: Indeed, in Figure 1 of the main manuscript we present the sample that shows mainly strong level signal stripe domains, grown in the **same batch** as all the samples presented in Figure 2. During our studies on this topic, we have investigated numerous SK/Fi/SK samples prepared nominally the same as the sample shown in this investigation. Figure R1 shows an example of another similar sample in which we actually did find hybrid weak and strong signals for the stripe domains (with varying ratios). These variations are most likely due to the (slightly) different magnetic properties of the Fi and/or SK layers, which can change in time due to our sputtering conditions for example. Since this point that the referee brings up is also related to tuning (the strong and weak) SK density, please also see our answer to the next point.

Figure R1: MFM images (all $4 \times 4 \mu\text{m}^2$) obtained on a similar SK/Fi/SK sample showing the coexistence of both weak and strong stripe domains and their field evolution into weak (incomplete) and strong (tubular) skyrmions.

2. The authors demonstrate the coexistence of two distinct skyrmion tubes in the same sample. However, I am feeling the density of strong skyrmion is much smaller than that of weak skyrmion. What is the reason? Is it possible to control the density of strong skyrmion?

Reply: To follow up with our answer above, the referee points out that the density of the strong (tubular) skyrmions is lower than that of the weak (incomplete) skyrmions, which is a correct observation. Our hypothesis is that the penetration of the strong skyrmions through the middle (Fi) layer depends on the properties of this Fi layer.

Our calculations in Supplementary Figure 3 of the supplementary material show that a tubular skyrmion can exist only if the Fi layer has a $|D| > 0.4\text{mJ/m}^2$. Together with the stability properties, a change in the D value of the Fi layer may have an influence on the skyrmion density and same arguments apply for the anisotropy and layer thickness.

Motivated partly by the observations shown in Figure R1, we have started new experiments with the goal of adjusting the density of the tubular skyrmions in a controlled way. Our preliminary MFM observations revealed that the density of the tubular skyrmions can be tuned by changing the anisotropy of the middle layer (this can be achieved - as discussed in the manuscript - by changing the Gd-to-Tb ratio) and by adjusting the thickness of this Fi layer. Moreover, we find that also varying the thickness of the ferromagnetic layers (i.e. Fe and Co) within the SK layers has a similar effect.

In the light of these preliminary results, we have included the following text in the manuscript:

'Since the SK/Fi/SK system presented here consists of multilayers with specifically tailored magnetic properties, it is possible that small variations in the properties of the individual layers could affect the tubular-to-incomplete skyrmion ratio. In fact, we already established that only particular values of D_{Fi} attributed to the Fi layer give rise to tubular skyrmions, and therefore one may speculate that additional parameters, such as Fi anisotropy and layer thickness, could play a role in determining this ratio. Also, regarding the skyrmion generator layer, [Ir(1)/Fe(0.3)/Co(0.6)/Pt(1)] \times 5, it was already established that changing the thickness of the ferromagnetic layers has a substantial influence on the skyrmion density⁶. This could be an alternative route to further tune the density of the two types of skyrmions.'

However, this study will be a direction for the design and realization of a next series of experiments with accompanying simulations.

3. By increasing the out-of-plane magnetic field, the authors demonstrate the conversion from strong skyrmion to weak skyrmion. I am thinking the weak skyrmion may not collapse suddenly at a critical field when further increasing the magnetic field. Instead, it may convert to a weaker one before its annihilation due to the field-induced switching of sub-SK layers. I suggest the authors discuss this problem and I would like to see more MFM data at larger magnetic field.

Reply: The referee brings up a very interesting point. The annihilation of skyrmions in such multilayer materials is certainly of high interest. Unfortunately, it is not trivial to unambiguously answer these questions, at least from an experimental point of view. This becomes apparent from the zoomed views (from Figure 1f-h in the main text) shown in in Figure R2. There, the data are displayed with a 1Hz (instead of 1.4Hz) frequency shift scale to make the weak-contrast skyrmions (highlighted by the small circles) better visible (the strong-contrast skyrmions are highlighted by the larger diameter circles). When increasing the field from 147mT (panel a) to 161mT (panel b) to 177mT (panel c), the contrast of the skyrmions generally becomes weaker. This is

Figure R2: Zoomed views of the MFM data displayed in Figure 1f-h in the main manuscript, but with a frequency shift scale of 1Hz to highlight the weak-contrast skyrmions (small-diameter circles), among the strong-contrast skyrmions (large-diameter circles). The contrast of weak skyrmions becomes smaller as the field is increased from 147mT (panel a) to 161mT (panel b). The contrast continues to decrease, when the field is increased to 177mT, and the top-right weak skyrmion (see arrow in panel c) has presumably been annihilated or may still exist in a form giving an MFM contrast that is too weak to permit any reasonable conclusion of its state. An investigation of a skyrmion annihilation process inside an SK layer and in its sublayers would clearly require a different experimental approach.

compatible with the expected shrinking of their radius in an applied field that is anti-parallel to their core magnetization, but could also arise from a shrinking of the length of a tubular skyrmion in the SK layers, i.e. by the annihilation of the skyrmion in an isolated Fe/Co layer. In fact, the referee is perfectly right when pointing out that the weak skyrmion does not collapse suddenly. We have performed micromagnetic simulations where the external field was steadily increased in steps of 2 mT. Figure R3 shows the results for a few selected field values. After each field step, the system was relaxed to equilibrium. Indeed, in the bottom SK layer the number of skyrmions is decreasing with the applied field until only two skyrmions are remaining in a field of about 162 mT.

Figure R3: Cross-sectional views for the SK/IL/SK sample, simulated in different (increasing) applied magnetic fields using magnum.af micromagnetic solver.

The stray field arising from a skyrmion running through all Fe/Co layers of the SK layer and a skyrmion that exists only in parts of the SK sub-layers is presumably very similar (particularly at the tip-to-magnetic layer distance selected in our experiment - 6nm oxidation protection layer and 15nm tip-sample distance). The stray fields for different micromagnetic skyrmion configurations at such a distance above the top-most magnetic layer of the sample will thus not be significantly different to allow a clear distinction between the different cases.

In conclusion, the MFM data we acquired on the samples presented in this manuscript do not allow further conclusions on the annihilation of the weak-contrast skyrmions, but would require a different experimental strategy involving dedicated samples and the use of the smallest possible tip-sample distance in the MFM experiments. Moreover, MFM tips with even smaller magnetic moments are necessary to minimize the tip influence on the skyrmion annihilation process and extensive modeling work to match candidate skyrmion magnetization structures to measured MFM data.

4. Related to my above comment, the authors only study the relaxed spin configurations of the strong and weak skyrmion tubes. I believe the authors are able to perform a more complex but useful simulation for the purpose of understanding the field-induced transformation between strong skyrmion and weak skyrmion. Namely, I am interested in what will happen if a larger out-of-plane magnetic field is applied to the relaxed state given in Fig. 3f. I expect it will convert to the state given in Fig. 3c but with slightly smaller skyrmion diameter. However, there is a possibility that the magnetic field switches the sub-SK layers instead of the Fi layer.

Reply: We thank the referee for stimulating us in performing a more detailed numerical study. By following his/her comment, we carried out additional micromagnetic simulations having the tubular skyrmion as initial state (Figure 3f in the main text, Figure R4a here) and where we increased the applied external field. Figure R4 shows the cross-sectional view of the conversion from the tubular skyrmion to the incomplete skyrmion for $\mu_0 H_{\text{ext}}=150$ mT at different time instants. The conversion process occurs via a gradual reduction of the skyrmion diameter, followed by the annihilation of the skyrmion in the Fi layer (Figure R4b), but we also observe the subsequent annihilation of the skyrmion in the first layer B1 of the bottom SK layer (Figure R4c). For fields lower than this value, the conversion does not occur, but the role of the field is only to reduce the tubular skyrmion diameter. We can conclude that, as expected, the external field drives the tubular to incomplete skyrmion conversion accompanied by a reduction of the diameter.

We have included Figure R4 together with corresponding text as Supplementary Note 6 in the supplementary material.

5. How do the authors make sure the weak skyrmion tubes are always coupled through the Fi layer? Why there is no decoupled weak skyrmion? Also, why there is no half weak skyrmion that only exists in the top or bottom SK layer? Are these problems related to the strength of magnetic field?

Reply: The referee is pointing out an important question. Our contrast data plotted in Figure 2o shows that the weak skyrmions have an average contrast of about 0.58Hz. The data further shows a decay of the skyrmion contrast with increased numbers of

Figure R4: Cross-sectional view of the tubular to incomplete skyrmion conversion when $\mu_0 H_{\text{ext}} = 150$ mT and for different time instants **a** 0 ns, **b** 0.8 ns, **c** 50 ns. We note that the annihilation of the skyrmion in the Fi layer happens at slightly different field strengths compared to the results shown in Figure R3 [i.e. PETASPIN (shown here) vs. magnum.af (Figure R3) micromagnetic solvers]. This can be easily due to the rapid application of the field with PETASPIN, which can induce a switching at lower fields.

fitted skyrmions, where our fitting routine starts with the weak skyrmion showing the strongest local contrast and then continues with the second strongest and so on. For this reason, the fitted contrast decays with increasing numbers of weak skyrmions. In order to fit the same number of strong and weak skyrmions we decided to fit only 24 weak skyrmions for Figure 2o. We did however also perform fits for all the weak skyrmions in the image and this is shown in Figure R5 below. These fits revealed that from the 103 weak skyrmions fitted in the image acquired at 146mT, only 8 show a contrast that is even weaker than the weak ones discussed in our present manuscript. This suggests that these very few “even-weaker” skyrmions may only be present in one of the SK multilayers (top one most likely). However, the number of such very weak skyrmions is small and their existence does not change the physics discussed in our manuscript.

We have added Figure R5 together with corresponding text as Supplementary note 5 in the supplementary material.

6. I do not understand why the authors only consider the interlayer exchange interaction between the Fi layer and B5 layer. What makes B5 layer different from T1 layer? Pt and Ir interfaces?

Reply: The referee raises an important question. The bottom Ir/Fe/Co/Pt layer finishes with a 1nm-thick Pt layer that is known to lead to a large RKKY exchange coupling to the Fi layer considered in our modeling work. The top SK layer grown on the top of the Fi layer starts with a 1nm-thick Ir layer. The RKKY coupling of 1nm Ir is very weak and, for this reason, it has been neglected here.

Figure R5: **a** Skyrmions at 147mT from Figure 1f in the main manuscript, but displayed with a contrast scale of 0.9Hz to make the weak skyrmions better visible. Note: Figure 1f is displayed (as all other panels in Figure 1) with 1.4Hz contrast scale. **b** Same image as in **a**, but with added crosses: red crosses indicate the positions of the 23 skyrmions showing a strong contrast that have been evaluated in Figure 2j and **c**; blue crosses indicate the 103 skyrmions showing a weak (thicker blue crosses) or very weak (fainter blue crosses). **c** The results for all the fitted skyrmions from **b**. As in Figure 2j, the contrast of the 23 strong skyrmions is noticeably different from that of the 103 weak skyrmions. Only 8 of the total 103 weak skyrmions have a contrast that is compatible with the skyrmions found in the single SK layer sample grown on the top of the Fi layer (see schematics in Figure 2b and MFM image in Figure 2g of the main manuscript).

7. According to the simulation results, the chirality and helicity of sub-layer skyrmions are not uniform due to the strong magnetostatic interaction. As the current-induced skyrmion dynamics is jointly determined by the skyrmion structure (e.g. helicity), I am unsure the weak and strong skyrmion tubes could be dynamic stable when spin-orbit torques are applied to drive them. I suggest the authors comment on this point and emphasize all advantages of skyrmion tubes in the main text.

Reply: Stimulated by this comment of the referee, we have performed preliminary micromagnetic simulations on the stability of the two skyrmions under a dc current. We have included the damping-like torque due to the SHE to the LLG equation considering a current density flowing in the Pt layers and the Fi layer (the Fi has a larger thickness compared to the ferromagnetic layer). We have considered a Gilbert damping of 0.1 and a spin-Hall angle of 0.1 [L. Liu *et al.*, Phys. Rev. Lett. 109, 096602 (2012)]. While the weak skyrmion follows the dynamical behavior of the Néel skyrmion in a single ferromagnetic layer, the strong skyrmion is stable for a smaller current range. As the damping-like torque applied to the Fi layer increases, the Bloch-like skyrmion stabilized there is annihilated because of the smaller saturation magnetization and anisotropy. In addition, considering that the thickness of the Fi layer is larger than the ferromagnetic ones, part of the current will also flow in the Fi layer, giving rise to a Zhang-Li torque:

$$\tau_{\text{ZL}} = \frac{\mu_B P}{eM_s} [(\mathbf{j}_{\text{Fi}} \cdot \nabla) \mathbf{m} - \beta \mathbf{m} \times (\mathbf{j}_{\text{Fi}} \cdot \nabla) \mathbf{m}], \quad (\text{R1})$$

where μ_B is the Bohr magneton, P is the polarization coefficient of the in-plane electrical current, \mathbf{j}_{Fi} is the electrical current flowing through the ferrimagnet, e is the electron charge, M_s is the effective ferrimagnet saturation magnetization, and β is the spin-torque non-adiabatic factor.

Due to the combination of the different chirality of the weak and strong skyrmions, together with different driving forces originated by the current for the two types of skyrmions (see Table R1), we expect that the dynamical properties exhibit a different behavior, however calling for a complete and systematic study which is beyond the scope of this work.

Type of Skyrmion	Spin-orbit torque	Zhang-Li torque
Weak skyrmion	Pt/Fe/Co/Ir	0 (A negligible current flow in the Ferromagnets and no skyrmion in the Fi layer)
Strong skyrmion	Pt/Fe/Co/Ir and Pt/Fi	Due to the current flowing in the Fi layer

Table R1: summary of the torques acting on the weak and strong skyrmion, respectively.

Minor Comments

1. Line 16: the racetrack-type memory device was first proposed by Parkin et al. as a domain-wall-based application. I suggest the authors cite the seminal paper on race-track memory.

Reply: Thank you for pointing this out. We have added this paper to the reference list.

2. Line 20: regarding the antiferromagnetically coupled skyrmion in synthetic antiferromagnets, I suggest the authors cite [Nature Communications 7, 10293 (2016)] and [Nature Communications 10, 5153 (2019)] along with Ref. 8. The suppression of skyrmion Hall effect was studied in the two papers, while Ref. 8 mainly focuses on the static stabilization of nanoscale synthetic antiferromagnetic skyrmions.

Reply: We have also added these papers in our reference list.

3. Line 34: At the beginning of the introduction, I think it would be better to mention the first theoretical prediction of the existence of skyrmions in magnets and cite the seminal paper by Bogdanov et al.

Reply: Thank you, we have added this paper also.

4. Line 45: The simultaneous existence of different skyrmion states can also be realized in frustrated magnets, as numerically shown in [Nature Communications 6, 8275 (2015); Nature Communications 8, 1717 (2017); Nature Communications 8, 14394 (2017)]. Besides, strictly speaking the chiral bobber is a three-dimensional structure different to the skyrmion tube. It is actually not a skyrmion-like object, so it is incorrect to say it a skyrmion bobber.

Reply: We have also added these suggested references. In addition, the referee is right that we were not very careful in our language when using 'skyrmion bobber'; we have changed it to 'chiral bobber'.

5. Line 46: Please spell out MBE.

Reply: We have done this.

6. Line 129: Fig. 1f should be Fig. 2f.

Reply: Thank you, we have changed it.

7. Line 148: Please spell out 3D.

Reply: We have done this.

8. Line 184: $D > 0$ should be $D < 0$.

Reply: Thank you for this comment. When reading that line again, we realized that it is a bit confusing and therefore changed the text in parentheses to 'see also Supplementary Fig. S4 in Supplementary Note 3 and compare the skyrmion diameter for positive and negative values of D_{FI} '.

9. Line 242: A full stop is missing after the equation.

Reply: We have added the period after the equation.

10. Figure 1: the orange square is not easily visible; I suggest the authors change the color to green to increase the readability. Figure 1: Please add color bar for the MFM data. Figure 2: The 'skyrmion number' usually means the topological charge of a skyrmion in the skyrmion community. How about change it to 'number of skyrmions'? Figure 3: It would be better to indicate the skyrmion diameter or simulated sample size.

Reply:

Figure 1: We have changed the square to green; we have also added in the caption that the MFM df range is 1.4Hz for all images (we chose not to add the color bar in Figure 1 as we do in Figure 2 due to space issues – we would prefer not to decrease the size of the sub-images that make up Figure 1).

Figure 2: Thank you for pointing this out; we have made the change to 'number of skyrmions'.

Figure 3: We have added scale bars to all the simulation figures.

Reviewer #2 (Remarks to the Author):

Mandru et al., introduce a new material architecture to stabilize magnetic skyrmions at room temperature.

Namely, a hybrid ferro/ferri/ferromagnetic multilayer which they argue can host two distinct skyrmion phases. Key to the coexistence of two skyrmion phases is the ferri-magnetic layer.

The purpose of their material is for the skyrmions generated by the top and bottom skyrmion layers to enter the ferrimagnetic layer. Thus, stabilize skyrmion-like tubular structures which penetrate through all layers of the sample.

Central to the study is also magnetic force microscopy measurements and associated quantitative analysis, which the team of Han Hug developed over the years.

Indeed, two magnetic force microscopy “contrasts” are observed with distinct field dependence. Following tests on several tailored samples, data suggest that one of two contrasts in magnetic force microscopy images arises from skyrmions existing solely in the bottom and top skyrmion layers, whereas the other is caused by tubular skyrmions running through all layers.

Experiments are complemented by micromagnetic simulations, giving details on the size and chirality of the textures across the multilayer. The simulations add credence to the proposal that the hybrid ferro/ferri/ferromagnetic multilayer favors the coexistence of two skyrmion phases at room temperature. That is, skyrmions at the top and bottom ferromagnetic layers and tubular skyrmions running through the entire sample.

Importantly, their material structure allows tunability of the skyrmion textures namely, tubular skyrmions can be converted into partial skyrmions.

I enjoyed reading this article. It is well written and easy to follow. The authors present a logical sequence of rationale and implementation. Their material structure introduces a novel tool to engineer (perhaps multiple) chiral textures.

The measurements are of high quality and the analysis of magnetic force microscopy is carefully done. I believe the authors know very well the limitations of magnetic force microscopy in elucidating details on skyrmion textures. The numerical studies accompanying the experiments offer the much needed support towards the key find of this work. Namely, two distinct type of skyrmion structures across the multilayer.

Although one may challenge the details of the interpretation provided by the authors, the experimental data are indicative of two magnetic textures.

Also, based on earlier studies on similar materials these textures must be magnetic skyrmions. The “contrast” characteristics the authors report can in fact be seen in other multilayers though not appreciated by the community before.

Hence, at the very least the authors deliver a new material and measurement platform and protocol to develop and study the coexistence of distinct “surface” and “bulk” magnetic skyrmions.

The simulations they report are realistic and within the norm and the results in tandem with the microscopy experiments. They help develop an interesting and potentially very useful protocol.

The work certainly encourages further studies in this field of research with promising technological impact as already mentioned their article.

I recommend publication.

Reply: We thank the reviewer very much for the positive comments regarding our combined experimental and theoretical study, particularly about the quality of the measurements and the much-needed simulations to complement these measurements. We are also very pleased that the reviewer enjoyed reading our manuscript and appreciated its possible impact.

Reviewers' Comments:

Reviewer #1:

Remarks to the Author:

Mandru and coworkers have carefully revised their manuscript and supplementary information. They have also adequately addressed the comments given in my previous report. I appreciate the authors' effort to perform additional experimental analysis and simulations. Now I can recommend the publication of this manuscript. Indeed, I still have some minor comments for the authors. I think the manuscript can be published after a minor revision and no more review is needed.

1. Regarding the authors' reply to my major comment #6, I suggest the authors provide some experimental references in the main text where appropriate. On page 18 of the revised manuscript, the authors write that the B5 layer is coupled to the first 1 nm of the ferrimagnetic layer via an RKKY-like interlayer exchange coupling and cite [43]. However, the reference 43 is a computational work.
2. While reading the supplementary information again, I noticed the sharp change of the skyrmion diameter at the T1/Fi interface in supplementary Fig. 3, which I think should be discussed in the supplementary note 3. This is, again, related to my major comment #6.
3. I suggest the authors add a color bar in Fig. 3. Also, please try to increase the font size in figures.

Reviewer #2:

Remarks to the Author:

I studied the referees' reports (one is mine), the authors' replies and the revised manuscript package. Also, I compared the revised manuscript with the original submission. The comments by reviewer#1 and the corresponding reply by the authors lend credence to my original understanding and judgement of the article's aims, contents and conclusions. I recommend publication.

Reply letter for Nature Communications manuscript NCOMMS-20-29411A

Reviewer Comments:

Reviewer #1 (Remarks to the Author):

Mandru and coworkers have carefully revised their manuscript and supplementary information. They have also adequately addressed the comments given in my previous report. I appreciate the authors' effort to perform additional experimental analysis and simulations. Now I can recommend the publication of this manuscript. Indeed, I still have some minor comments for the authors. I think the manuscript can be published after a minor revision and no more review is needed.

Reply: Again, we thank the reviewer very much for the positive assessment of our work and for the suggestions for improvement. Below we address in detail each comment point by point.

1. Regarding the authors' reply to my major comment #6, I suggest the authors provide some experimental references in the main text where appropriate. On page 18 of the revised manuscript, the authors write that the B5 layer is coupled to the first 1 nm of the ferrimagnetic layer via an RKKY-like interlayer exchange coupling and cite [43]. However, the reference 43 is a computational work.

Reply: Thank you for pointing out this issue. **Experimental references (for coupling through Pt and Ir) are now present in the same section as reference 43.** Additionally, in order to better explain the (different) couplings at the two interfaces, **we added the following text** (similar to our previous answer to major comment #6 of the referee) **in the same section**:

'The bottom Ir/Fe/Co/Pt layer finishes with a 1nm-thick Pt layer that is known to lead to a large RKKY exchange coupling to the Fi layer; we set a positive value of the constant (ferromagnetic coupling) equal to 0.8mJ/m² from ⁴⁴. The top SK layer grown on the top of the Fi layer starts with a 1nm-thick Ir layer. Since the RKKY coupling through 1nm of Ir is known to be very weak ⁴⁵, it has been neglected here'.

2. While reading the supplementary information again, I noticed the sharp change of the skyrmion diameter at the T1/Fi interface in supplementary Fig. 3, which I think should be discussed in the supplementary note 3. This is, again, related to my major comment #6.

Reply: Indeed, as the referee points out, the change in diameter at the Fi/T1 interface has to do with the coupling through Ir, which is very weak compared to the coupling through Pt (for the B5/Fi case, where the spins are locked and the diameter is the same as in the Fi). In order to address this point, **we have added the following discussion in Supplementary note 3**:

'We note the difference in skyrmion diameter between B5/Fi and Fi/T1 interfaces. This is due to the fact that the skyrmions at the lower interface are strongly coupled to the Fi as a result of the large exchange coupling through the 1 nm of Pt. Since the skyrmions in the top layer are very weakly coupled

through the 1 nm of Ir with the Fi, their diameter depends mainly on the SK layer parameters and has therefore a different equilibrium size'.

3. I suggest the authors add a color bar in Fig. 3. Also, please try to increase the font size in figures.

Reply: A color bar is already present in Fig. 3, right above the smaller 50nm scale bar. We have also increased the font size in all figures wherever possible.

Reviewer #2 (Remarks to the Author):

I studied the referees' reports (one is mine), the authors' replies and the revised manuscript package. Also, I compared the revised manuscript with the original submission. The comments by reviewer#1 and the corresponding reply by the authors lend credence to my original understanding and judgement of the article's aims, contents and conclusions. I recommend publication.

Reply: We again thank the reviewer very much for the positive comments regarding our combined experimental and theoretical study.